# Label-Aware Global Consistency for Multi-Label Learning with Single Positive Labels

**Ming-Kun Xie**,* **Jia-Hao Xiao**,* **Sheng-Jun Huang**†

College of Computer Science and Technology, Nanjing University of Aeronautics and Astronautics
MIIT Key Laboratory of Pattern Analysis and Machine Intelligence, Nanjing, 211106
{mkxie,jiahaoxiao,huangsj}@nuaa.edu.cn

## Abstract

In single positive multi-label learning (SPML), only one of multiple positive labels is observed for each instance. The previous work trains the model by simply treating unobserved labels as negative ones, and designs the regularization to constrain the number of expected positive labels. However, in many real-world scenarios, the true number of positive labels is unavailable, making such methods less applicable. In this paper, we propose to solve SPML problems by designing a Label-Aware global Consistency (LAC) regularization, which leverages the manifold structure information to enhance the recovery of potential positive labels. On one hand, we first perform pseudo-labeling for each unobserved label based on its prediction probability. The consistency regularization is then imposed on model outputs to balance the fitting of identified labels and exploring of potential positive labels. On the other hand, by enforcing label-wise embeddings to maintain global consistency, LAC loss encourages the model to learn more distinctive representations, which is beneficial for recovering the information of potential positive labels. Experiments on multiple benchmark datasets validate that the proposed method can achieve state-of-the-art performance for solving SPML tasks.

## 1 Introduction

Multi-label learning (MLL) is a practical and effective learning framework for tackling objects with complex semantics, where each instance is assumed to be associated with multiple class labels. For example, an image may be annotated with labels *beach*, *sea* and *sky* simultaneously. Multi-label learning aims to train a classifier that can accurately predict all the relevant labels for unseen instances. It has been successfully applied into many real-world applications, such as image annotation [2], scene understanding [22], facial attribute recognition [13].

Given the output space is exponentially larger than that of single-label learning, it often requires a large number of examples with precise annotations to train an effective MLL classifier. However, in many real-world scenarios, it is difficult and costly to collect the precise annotations. To reduce the labeling cost, an alternative solution is to ask the annotators to assign the most obvious label to each instance, leading to only a single positive label available. Such problem has been formulated as a learning framework called single positive multi-label learning (SPML) [4]. In this case, except for the annotated positive label, each of other labels could be positive or negative but unknown, thus the learning task becomes much more challenging due to the lack of supervision.

To solve SPML problems, the previous work [4] intuitively designs the "assume negative" (AN) loss, which treats all unobserved labels as negative ones, and trains an MLL classifier with the conventional

---

*Both authors contributed equally to this research.
†Correspondence to: Sheng-Jun Huang (huangsj@nuaa.edu.cn).

36th Conference on Neural Information Processing Systems (NeurIPS 2022).

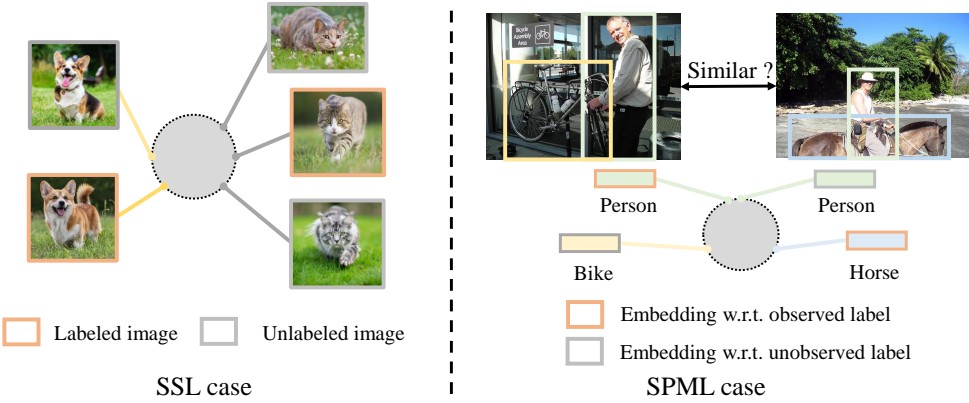

Figure 1: An illustration of the comparison between clustering regularization and our proposed label-aware global consistency regularization.

binary cross entropy (BCE) loss. Unfortunately, AN loss often introduces many false negative labels, leading to a noticeable decrease in performance of the model. To alleviate the negative impact of false negative labels, authors further propose a regularization to constrain the number of expected positive labels. However, since the true number of positive labels cannot be accessible in many real-world scenarios, it often requires additional estimation or elaborate hyper-parameter tuning to determine its true value.

Clustering assumption [35] is first proposed to solve semi-supervised learning (SSL) problems. It aims to recover the true labels for unlabeled examples by utilizing the global consistency, *i.e.*, encouraging points on the same structure (typically referred to as a clustering) to be assigned with the same label. The idea of clustering assumption has been successfully extended to solve many weakly-supervised learning problems [23, 28]. Unfortunately, applying clustering regularization to solve SPML problems is not as straightforward as in SSL case, since it is difficult to measure the underlying similarity between any two examples with multiple labels. For example, in SSL case (see the left side in Figure 1), it is feasible to distinguish examples from different classes by measuring their similarity based on image-level feature representations. While in multi-label case (see the right side in Figure 1), it is hard to measure the similarity between any two images, which share the same object (*e.g.*, person), while still have individual objects (*e.g.*, bike and horse).

In this paper, we design a Label-Aware global Consistency (LAC) regularization to leverage the manifold structure information for solving SPML problems. On one hand, we first identify the potential positive labels mislabeled by AN loss based on class probabilities predicted by the model. The pseudo-labeling consistency regularization is then used for training model to boost its ability on identifying the potential positive labels. On the other hand, as illustrated in the right side of Figure 1, the LAC loss is designed to encourage the global consistency among label-wise embeddings, *i.e.*, pulling together intra-class embeddings while pulling apart inter-class embeddings. By leveraging the manifold structure information, the LAC loss enhances the distinctiveness of learned feature representations, which is beneficial for identifying the potential positive labels. Extensive experimental results on multiple benchmark datasets demonstrate that our method can achieve state-of-the-art performance.

## 2 Related Work

There are a plenty of studies in multi-label learning literature. Existing methods can be roughly classified into three groups. Considering the traditional binary cross entropy (BCE) loss often suffers from the positive-negative imbalance issue, the first kind of methods aims to design specific loss functions to alleviate this issue. Among them, asymmetric loss (ASL) [11] is a representative method that dynamically down-weights and hard-thresholds easy negative examples. The second kind of methods is proposed to capture the label correlations, which are regarded as an essential element for improving the practical performance of MLL. ML-GCN [2] and its variant [30] utilize the graph convolution network (GCN) to capture the co-occurrence correlations among labels. Considering an

image often contains multiple objects, the last kind of methods is designed to locate areas of interest related to semantic labels by using attention technique.

Traditional multi-label learning assumes that each instance has been annotated with all of its relevant labels. However, in many real-world scenarios, it is costly and difficult to precisely assign an image with multiple labels. To solve this issue, the current efforts are mainly dedicated to reducing the labeling cost by training an MLL model with only partial labels. For example, [9] fine-tune the pre-trained model on ImageNet with partial annotations and perform pseudo-labeling to recover the potential positive labels. In [16], authors propose a two-stage interactive learning framework that performs similarity learning and CNN training interactively to improve performance of each other. [20] propose an unified framework to exploit instance-level and prototype-level semantic representation for recovering the potential positive labels.

SPML is the extreme case of multi-label learning with partial labels, where only one of multiple potential positive labels can be observed. The earliest work intuitively treats all unobserved labels as negative ones, and trains an MLL model by introducing the regularization terms that alleviate the negative impact of false negative labels [4]. The spatial consistency [26] is designed to avoid the bias toward negative prediction by maintaining the consistency of classification maps between the network and exponential moving average (EMA) model. Unlike aforementioned methods, [36] treat unobserved labels as unknown and propose to maximize the entropy of predicted probabilities for unobserved labels. Furthermore, authors combine the asymmetric pseudo-labeling and self-paced strategy to obtain more accurate pseudo labels.

Besides SPML, a variety of weakly-supervised multi-label learning frameworks have been widely studied, including learning with multi-label noise [34], partial multi-label learning [32, 24], few-shot multi-label learning [1], learning with pairwise relevance comparison [33], and semi-supervised multi-label learning [29].

## 3   The Proposed Method

In the SPML problem, let $x \in \mathcal{X}$ be a feature vector and $\hat{y} \in \mathcal{Y}$ be its corresponding labels, where $\mathcal{X} = \mathbb{R}^d$ is the feature space and $\mathcal{Y} = \{0, 1\}^q$ is the target space with $q$ possible class labels. Here, $\hat{y}_j = 1$ indicates the $j$-th label is the only observed positive label for instance $x$ while $\hat{y}_j = 0$ indicates the $j$-th label cannot be observed. Since only a single positive label can be observed, we have $\sum_{j=1}^{q} \hat{y}_{ij} = 1$ for every instance $x_i$. We further denote by $y$ the true label vector for instance $x$. Let $p(y|x)$ be the predicted probability distribution over classes and $p(y_j|x)$ be the predicted probability of the $j$-th class for input $x$. We use $[q]$ to denote the integer set $\{1, ..., q\}$.

**Label-wise Embedding Model**   To obtain the label-wise embedding with respect to every label, existing methods can be roughly divided into two groups, *i.e.*, global-average-pooling (GAP) based methods [31, 12] and attention-based methods [18, 21, 6]. For simplicity, we adopt the attention-based method to construct the label-wise embedding model . Assume that the label-wise embedding model is composed of a label-wise embedding decoder $f$, which can generate a high-dimensional label-wise feature representation $f_j(x)$ with respect to the $j$-th label of instance $x$, and a classification head $h$, which can output the predicted probability of the $j$-th class $p(y_j|x) = h(f_j(x))$. We further denote by $g(\cdot)$ a non-linear projection head, which transforms a high-dimensional embedding $f_j(x)$ to a low-dimensional embedding $z_j = g(f_j(x))$.

In traditional multi-label classification, the most commonly used loss function is binary cross entropy (BCE) loss, which decomposes the original task into multiple binary classification problems. Formally, given a batch of examples $\{(x_i, y_i)\}_{i=1}^{b}$, the BCE loss can be defined as follows:

$$\mathcal{L}_{\text{BCE}} = -\frac{1}{b} \sum_{i=1}^{b} \sum_{j=1}^{q} y_{ij} \log(p(y_j|x_i)) + (1 - y_{ij}) \log(1 - p(y_j|x_i)) \tag{1}$$

**The assume negative (AN) loss**   However, the conventional BCE loss cannot be directly applied to solve SPML problems, where only a single positive label can be accessible for every instance. Training the neural network with only one positive label often makes model collapse to a trivial solution. To mitigate this issue, an intuitive method is to treat all unobserved labels as negative ones, and then the neural network can be trained with the resulting AN loss [4] given a batch of SPML

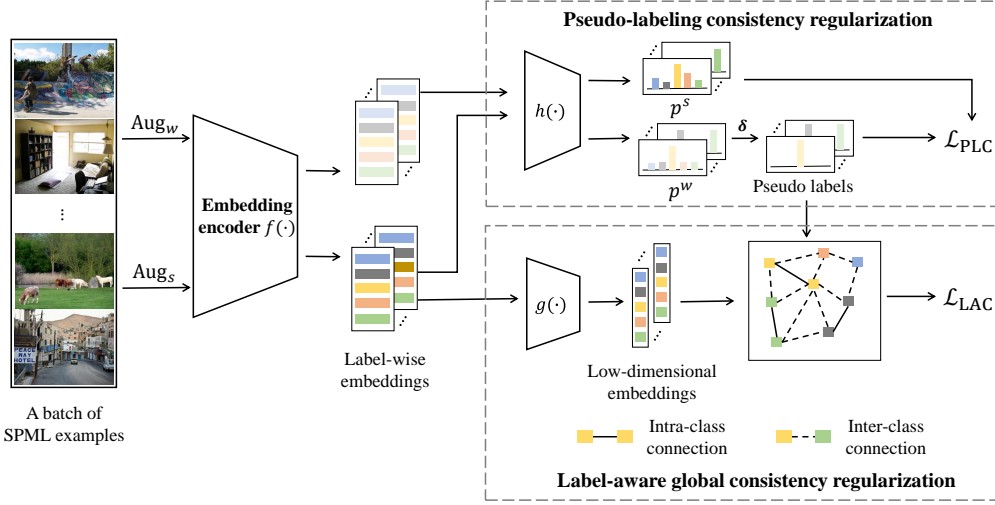

Figure 2: An illustration of the proposed learning framework.

examples $\{(\boldsymbol{x}_i, \hat{\boldsymbol{y}}_i)\}_{i=1}^b$:

$$\mathcal{L}_{\text{AN}} = -\frac{1}{b} \sum_{i=1}^b \sum_{j=1}^q \hat{y}_{ij} \log(p(y_j|\boldsymbol{x}_i))) + (1 - \hat{y}_{ij}) \log(1 - p(y_j|\boldsymbol{x}_i))) \tag{2}$$

Unfortunately, AN loss often introduces a large number of false negative labels, which leads to a noticeable decrease in generalization performance of the model. To alleviate the negative impact of false negatives, we design the following two regularization losses: 1) the pseudo-labeling consistency regularization $\mathcal{L}_{\text{PLC}}$ to identify the potential positive labels mislabeled by AN loss, 2) the label-aware global consistency loss $\mathcal{L}_{\text{LAC}}$ to leverage the manifold structure information for learning a more distinctive feature representation. The overall objective function can be defined as:

$$\mathcal{L} = \mathcal{L}_{\text{AN}} + \lambda_{\text{PLC}}\mathcal{L}_{\text{PLC}} + \lambda_{\text{LAC}}\mathcal{L}_{\text{LAC}} \tag{3}$$

Figure 2 provides an illustration of the proposed learning framework. For a batch of SPML examples, we generate the pseudo labels based on their weakly-augmented versions. On one hand, the consistency regularization is conducted between the pseudo labels and model outputs on the strongly-augmented versions to prevent from over-fitting to false negative labels. On the other hand, the label-aware global consistency is used to learn more distinctive feature representations such that intra-class embeddings are pushed together while inter-class embeddings are pushed away. In the following content, we will introduce these two regularization losses in detail.

### 3.1 Pseudo-Labeling Consistency Regularization

In this section, we propose the pseudo-labeling consistency regularization for recovering the ground-truth labeling information of unobserved labels.

We define the loss $\mathcal{L}_{\text{PLC}}$ between the pseudo labels and the model predictions. Specifically, for the $j$-th label of $\boldsymbol{x}_i$, let $p_{ij}^w = p(y_j|\text{Aug}_w(\boldsymbol{x}_i))$ denote the pseudo labels, where $\text{Aug}_w(\cdot)$ is a stochastic weak augmentation. To avoid the trivial solution, let $p_{ij}^s = p(y_j|\text{Aug}_s(\boldsymbol{x}_i))$ denote the model prediction, where $\text{Aug}_s(\cdot)$ is a stochastic strong augmentation. Then, the pseudo-labeling consistency loss can be formulated as:

$$\mathcal{L}_{\text{PLC}} = -\frac{1}{b} \sum_{i=1}^b \sum_{j=1}^q \mathbb{I}(p_{ij}^w \geq \delta | \hat{y}_{ij} = 0) \log(p_{ij}^s) + \mathbb{I}(p_{ij}^w \leq \bar{\delta} | \hat{y}_{ij} = 0) \log(1 - p_{ij}^s) \tag{4}$$

where $\delta$ (or $\bar{\delta} = 1 - \delta$) is a threshold that controls how many positive (or negative) pseudo labels would be used for model training. Here, $\mathbb{I}(\cdot)$ is the indicator function that output 1 if the condition

satisfies while 0, otherwise, and the condition $\hat{y}_{ij} = 0$ is used to ensure the loss $\mathcal{L}_{\text{PLC}}$ is only imposed on the unobserved labels. Generally, the loss $\mathcal{L}_{\text{PLC}}$ imposes a consistency regularization on the model such that encourages it to output the same probability with respect to an unobserved label for an example even after it is strongly augmented. By using a strong augmentation, $\mathcal{L}_{\text{PLC}}$ would alleviate the over-fitting issue to false negative labels and improve the ability of identifying potential positive labels.

## 3.2 Label-Aware Global Consistency Regularization

As discussed in the above section, $\mathcal{L}_{\text{PLC}}$ can be regarded as a consistency regularization on the high-level representation, *i.e.*, the final output of network. With the increase of the depth of the neural network, the impact on network parameters of enforcing the consistency on the high-level representation may be gradually degraded [12], and thus weakens the effectiveness of the proposed loss $\mathcal{L}_{\text{PLC}}$. To further alleviate the harmfulness caused by the false negative labels, motivated by the clustering assumption [35], we design label-aware global consistency regularization $\mathcal{L}_{\text{LAC}}$ to maintain global consistency among label-wise embeddings. The main idea is to pull intra-class label-wise embeddings to be close while pull inter-class label-wise embeddings to be separated. The intuition is that by maintaining the global consistency among label-wise embeddings, LAC enhances the distinctiveness of learned representations, which is beneficial for recovering the true labeling information of potential positive labels.

To derive the $\mathcal{L}_{\text{LAC}}$ loss, for a batch of examples, we first use the label-wise embedding decoder $f$ and projection head $g$ to produce a set of embeddings $\mathcal{Z} = \{z_k | 1 \leq k \leq B\}$, where $B = b \cdot q$ is the total number of embeddings for $b$ instances with $q$ classes. For notational convenience, for the embedding $z_k$ with respect to $j$-class of instance $x_i$, we define the mapping $i = i(k), j = j(k)$ to bridge the embedding index $k$ with its original instance index $i$ and class index $j$.

In order to exploit as more as possible supervised information to learn distinctive representations, we compute the $\mathcal{L}_{\text{LAC}}$ loss not only on unobserved labels but also on observed ones. To obtain a unified formulation, for the single positive label $\hat{y}_{ij} = 1$, we treat its pseudo label as $p_{ij}^w = 1$ although it is actually the true label. For any two intra-class label-wise embeddings $z_k$ and $z_{k'}$, by denoting $i = i(k), j = j(k)$ and $i' = i(k'), j' = j(k')$, we can construct the intra-class connection matrix $I$ of size $B \times B$ to connect them:

$$I_{kk'} = \begin{cases} 1 & \text{if } j = j' \text{ and } p_{ij}^w \geq \delta, p_{i'j'}^w \geq \delta \\ 0 & \text{otherwise} \end{cases} . \tag{5}$$

From the equation Eq.(5), every embedding a pseudo label larger than the threshold $\delta$ would be treated as the anchor embedding and be connected with embeddings from the same class that satisfy the same condition. Since a low prediction probability indicates the corresponding label is likely to be negative, its embedding is invalid and would never be used.

Similarly, we can construct the inter-class connection matrix $\bar{I}$ of the same size to connect any two inter-class label-wise embeddings:

$$\bar{I}_{kk'} = \begin{cases} 1 & \text{if } j \neq j' \text{ and } p_{ij}^w \geq \delta, p_{i'j'}^w \geq \delta \\ 0 & \text{otherwise} \end{cases} . \tag{6}$$

From the equation Eq.(6), an anchor embedding is connected with embeddings from other classes with pseudo labels larger than a threshold $\delta$.

Next, we can train the label-wise embedding decoder $f$ and the projection head $g$ by enforcing the global consistency among label-wise embeddings. It is expected to pull together an anchor label-wise embedding and its intra-class label-wise embeddings while separate the anchor label-wise embedding from its inter-class label-wise embeddings. To achieve this goal, given the connection matrices $I$ and $\bar{I}$, we define label-aware global consistency loss as follows:

$$\mathcal{L}_{\text{LAC}} = -\frac{1}{B} \sum_{k=1}^{B} \sum_{k'=1}^{B} I_{kk'} \log \frac{\exp(z_k \cdot z_{k'}/\tau)}{\sum_{t=1}^{B} \bar{I}_{kt} \exp(z_k \cdot z_t/\tau)} \tag{7}$$

It is noteworthy that when $k = k'$, *i.e.*, an anchor embedding and itself, in order to avoid a trivial solution, we produce two different embeddings $z_k$ and $z_{k'}'$ by imposing two different augmentations to its original training image.

**Memory queue**. Inspired of the previous work [14], we introduce a memory queue to store the past $K$ embeddings and their corresponding pseudo labels $\text{MQ} = \{(z_k, p_k^w)|k \in [K]\}$, where $p_k^w$ is the pseudo label for embedding $z_k$. It is noteworthy that the memory queue consists of embeddings with respect to both unobserved and observed labels. Then, the connection matrices $I$ and $\bar{I}$ are extended to size of $B \times (B + K)$. This means we would impose the label-aware global consistency regularization between an anchor embedding of the current embedding set and every embedding of the memory queue.

**Two-stage training strategy**. In our experiments, we find that training model with Eq.(3) in an end-to-end fashion often leads to an unfavorable performance that can be worse than that without $\mathcal{L}_{\text{LAC}}$. The main reason is that at the early stage of training, the model suffers from the insufficient training issue, which leads model to produce low-quality label-wise embeddings. Enforcing low-quality label-wise embeddings to maintain global consistency may destroy the feature representation learning and thus significantly degrades model performance. To solve this issue, we propose a two-stage training strategy, which aims to let model learn a high-quality label-wise embeddings firstly and then perform LAC regularization. Specifically, at the first stage, we train the model by minimizing AN loss and PLC loss, *i.e.*, $\mathcal{L}_{\text{AN}} + \mathcal{L}_{\text{PLC}}$, until it reaches the convergence. The model is expected to produce sufficiently high-quality label-wise embeddings for the second stage training. At the second stage, we fine-tune the model with Eq.3 consisting of all three losses to learn a more distinctive representation.

## 4 Experiments

In this section, we first perform experiments to compare our method with state-of-the-art methods; then, we conduct ablation studies to examine the effectiveness of each component for the proposed method.

### 4.1 Experimental Settings

**Datasets.** We perform experiments to evaluate our proposed method on four benchmark datasets: Pascal VOC-2012 (VOC for short) [3] [10], MS-COCO-2014 (COCO for short) [4] [17], NUS-WIDE (NUS for short) [5] [3], and CUB-200-2011 (CUB for short) [6] [27]. Table 1 reports the detailed characteristics of four benchmark datasets. Specifically, VOC contains 5,717 training images and 5,823 validation images for 20 classes. We divide the training set into 4,574 training examples and 1,143 validation examples, and then use the original validation set for testing. COCO contains 82,081 training images and 40,137 validation images for 80 classes. We divide the training set into 65,665 training examples and 16,416 validation examples, and then use the original validation set for testing. NUS is not complete online because of many invalid URLs. We sent a request email to the authors and obtain the complete dataset, which consists of 126,034 training images and 84,226 testing images for 81 classes. Following [4], we merge all images and randomly select 150,000 training samples and 60,260 testing examples. Finally, we withhold 30,000 images from training examples for validation. CUB contains 5,994 training images and 5,794 testing images. Following [4], instead of using 200 bird categories as class labels, we assign a vector that indicates the presence or absence of 312 binary attributes to every image. We divide the original training set into 4,795 training examples and 1,199 validation examples. In order to compare our method with state-of-the-art methods, we use the code shared by [4] to generate the training, validation and testing sets. For each dataset, we withhold 20% of the training examples for validation. To construct SPML data, we randomly select one positive label for each training example, while keep the validation and testing sets always fully labeled. Follow [4], We report the mean average precision (mAP) on the testing set by using predictions of the model with the best validation performance.

**Comparing methods.** Besides the baseline **AN** loss, we compare the proposed method with the following state-of-the-art algorithms: **WAN** [4], which introduces a weight parameter to down-weight the losses with respect to negative labels; **EPR** [4], which utilizes an expected positive

---

[3]`http://host.robots.ox.ac.uk/pascal/VOC/`

[4]`https://cocodataset.org`

[5]`https://lms.comp.nus.edu.sg/wp-content/uploads/2019/research/nuswide/NUS-WIDE.html`

[6]`http://www.vision.caltech.edu/visipedia/CUB-200-2011.html`

Table 1: The detailed characteristics of benchmark datasets.

| Dataset | # Training | # Validation | # Testing | # Classes |
|---------|-----------|--------------|-----------|-----------|
| VOC | 4,574 | 1,143 | 5,823 | 20 |
| COCO | 65,665 | 16,416 | 40,137 | 80 |
| NUS | 120,000 | 30,000 | 60,260 | 81 |
| CUB | 4,795 | 1,199 | 5,794 | 312 |

regularization to mitigate the negative impact of false negative labels; **AN-LS**, which incorporates the label smoothing technique to AN loss; **ROLE** [4], which adopts regularized online label estimation (ROLE) technique to alleviate the negative impact of false negative labels; **EN+SCL** [26], which combines expected negative loss with the spatial consistency loss (SPL); **EM+APL** [36], which incorporates asymmetric pseudo-labeling (APL) loss into entropy minimization.

**Implementation.** The label-wise embedding model consists of two components, including a backbone for extracting visual features, and a label-wise embedding decoder for producing label-wise embeddings. Following [4], we use a ResNet-50 [15] pretrained on the ImageNet [7] as the backbone to extract features. Then, the extracted features are fed into the label-wise embedding decoder to produce label-wise embeddings. The label-wise embedding decoder consists of a standard self-attention block and a cross-attention block [25]. After obtaining label-wise embeddings, we can feed them into the classification head $h$ consisting of a linear layer to obtain the final predictions. By feeding label-wise embeddings into the projection head $g$ consisting of two linear layers and a ReLU layer, we can obtain the low-dimensional embeddings for performing label-aware global consistency regularization. In particular, the training process consists of two stages, and only the second stage training needs the low-dimensional embeddings produced by the projection head $g$. Therefore, we freeze the parameters of projection head at the first stage. We resize the resolution of input images to $448 \times 448$ as in [4]. For training images, we use both a weak augmentation (only containing random horizontal flipping) and a strong augmentation (containing Cutout [8] and RandAugment [5]). For training the model, we use the AdamW [19] optimizer with the weight decay of $0.01$. The OneCycleLR scheduler is used to change the learning rate with the max learning rate of $0.0001$. We train the model for 40 epochs with the early stopping. We consider the batch size in the range of $\{8, 16, 32, 64\}$. At the first stage, we set $\lambda_{\mathrm{PCL}} = 1$ and determine the threshold $\delta$ from the range of $\{0.5, 0.6, 0.7, 0.8, 0.9\}$. At the second stage, there are two extra parameters, including the balancing parameter $\lambda_{\mathrm{LAC}}$, the size of memory queue $K$. We set $\lambda_{\mathrm{LAC}} = 1$, and determine $K$ from the range of $\{512, 1024, 2048, 4096, 8192\}$, respectively. All hyperparameters are determined according to their mAP obtained on the validation set. In Section 4.4, we conduct experiments to analyze the sensitivity for unfixed parameters. Furthermore, we apply exponential moving average (EMA) to model parameters $\theta$ with a decay of 0.9997. We perform all experiments on GeForce RTX 3090 GPUs. The random seed is set to 1 for all experiments

## 4.2 Comparison Results with the State-of-the-Arts

Table 2 reports comparison results between the proposed method and comparing methods in terms of mAP on four benchmark datasets. To make a fair comparison, besides the results reported in their original paper (marked by ResNet-50), we still report the results of comparing methods trained with the label-wise embedding model (LEM) (marked by LEM). It is noteworthy that for EN methods, including $\mathcal{L}_{\mathrm{EN}}+\mathcal{L}_{\mathrm{CL}}$ and $\mathcal{L}_{\mathrm{EN}}+\mathcal{L}_{\mathrm{SCL}}$, in their original paper, only the results on VOC and COCO are reported. Furthermore, we cannot reproduce these two methods based on LEM due to the inaccessibility of source codes. From the table, it can be observed that: 1) AN loss achieves unfavorable performance in almost all cases. This validates that simply using AN cannot effectively solve SPML problems, since it introduces a large number of false negative labels, which are harmful for model training. 2) ROLE method cannot be easily adapted to the label-wise embedding model, since its performance suffers from a significant drop by replacing the original ResNet-50 with LEM. One possible reason is that we fail to perform "LinearInit." that has been used to initialize the weights of the classifier based on given feature representations in its original implementation, since these given feature representations cannot be used by LEM. 3) Our proposed method consistently achieves desirable performance and achieves the best performance on almost all cases except for CUB, where $\mathcal{L}_{\mathrm{EM}}+\mathcal{L}_{\mathrm{APL}}$ outperforms our method with minor improvement. In particular, our method achieves

Table 2: Mean average precision (mAP) of each comparing method on four benchmark datasets. The best performance is highlighted in bold.

| Methods | LEM | | | | ResNet-50 | | | |
|---|---|---|---|---|---|---|---|---|
| | VOC | COCO | NUS | CUB | VOC | COCO | NUS | CUB |
| $\mathcal{L}_{\text{WAN}}$ | 89.2 | 73.5 | 48.5 | 22.5 | 86.5 | 64.8 | 46.3 | 20.3 |
| $\mathcal{L}_{\text{EPR}}$ | 88.8 | 72.7 | 49.3 | 23.1 | 85.5 | 63.3 | 46.0 | 20.0 |
| $\mathcal{L}_{\text{AN}}$ | 87.6 | 72.3 | 48.5 | 18.6 | 85.1 | 64.1 | 42.0 | 19.1 |
| $\mathcal{L}_{\text{AN-LS}}$ | 88.0 | 70.9 | 47.1 | 16.3 | 86.5 | 69.2 | 50.5 | 16.6 |
| $\mathcal{L}_{\text{ROLE}}$ | 88.1 | 69.6 | 44.5 | 14.2 | 88.2 | 69.0 | 51.0 | 16.8 |
| $\mathcal{L}_{\text{EN}} + \mathcal{L}_{\text{CL}}$ | - | - | - | - | 87.6 | 71.6 | - | - |
| $\mathcal{L}_{\text{EN}} + \mathcal{L}_{\text{SCL}}$ | - | - | - | - | 88.0 | 72.1 | - | - |
| $\mathcal{L}_{\text{EM}}$ | 89.2 | 73.2 | 48.7 | 22.2 | 89.1 | 70.7 | 47.2 | 20.9 |
| $\mathcal{L}_{\text{EM}} + \mathcal{L}_{\text{APL}}$ | 89.2 | 73.1 | 48.6 | **23.6** | 89.2 | 70.9 | 47.6 | 21.8 |
| Ours | **89.6** | **75.6** | **51.1** | 23.3 | - | - | - | - |

Table 3: The ablation studies on VOC and COCO.

| Methods | | | | | mAP (%) | |
|---|---|---|---|---|---|---|
| Assume negative | PLC | LAC | Two-stage | Memory queue | VOC | COCO |
| √ | | | | | 87.6 | 72.3 |
| √ | √ | | | | 89.0 | 73.7 |
| √ | √ | √ | | | 84.5 | 65.6 |
| √ | √ | √ | √ | | 89.2 | 74.7 |
| √ | √ | √ | √ | √ | 89.6 | 75.6 |

favorable performance on two large-scale datasets and improve upon the state-of-the-art performance by $2.1\%$ on COCO and $1.8\%$ on NUS in terms of mAP. These results validate that the proposed method can effectively solve SPML problems.

### 4.3 Ablation Studies

In this section, to further analyze how the proposed method improves performance of SPML, we conduct a series of ablation studies on VOC and COCO and report the results on Table 3. We first conduct the experiment to validate the effectiveness for the proposed pseudo-labeling consistency regularization $\mathcal{L}_{\text{PLC}}$. The achieved performance is $89.0\%$ on VOC and $73.7\%$ on COCO by utilizing $\mathcal{L}_{\text{PLC}}$ that are better than $87.6\%$ and $72.3\%$ achieved by only utilizing AN loss $\mathcal{L}_{\text{AN}}$. This discloses that by using $\mathcal{L}_{\text{PLC}}$, the model can recover the true positive labels that benefits for model training. To validate the effectiveness of the proposed label-aware global consistency regularization LAC, as discussed in Section 3.2, we train the model with two strategies, including end-to-end training, which directly optimizes the joint loss Eq.(3) in a pipeline, and two-stage training, which first trains the model with the first two losses until the model converges, and then trains the model with Eq.(3). It is can be observed that by adopting the end-to-end training strategy, the performance is degraded from $89.0\%$ to $84.5\%$ on VOC and $73.7\%$ to $65.6\%$ on COCO by adding LAC into training while are improved to $89.2\%$ and $74.7\%$ by utilizing two-stage training. This is because by adopting the end-to-end training strategy, at the early stage of model training, it is difficult for the model to produce high-quality label-wise embeddings for performing LAC regularization. Enforcing low-quality label-wise embeddings to maintain the global consistency may destroy the feature representation learning and thus lead to a noticeable decrease in model performance. Compared to end-to-end training, the two-stage training strategy primarily learn high-quality label embeddings at the first stage, and then utilize the LAC regularization to encourage global consistency among label-wise embeddings, which improves the discrimination ability of the model. Finally, we examine the usefulness of memory queue. By using the memory queue, the performance is improved with $0.4\%$ increment from $89.2\%$ on VOC and with $0.9\%$ increment from $74.7\%$. The main reason is that the memory queue provides the model with much more supervision for learning more distinctive representations, which are

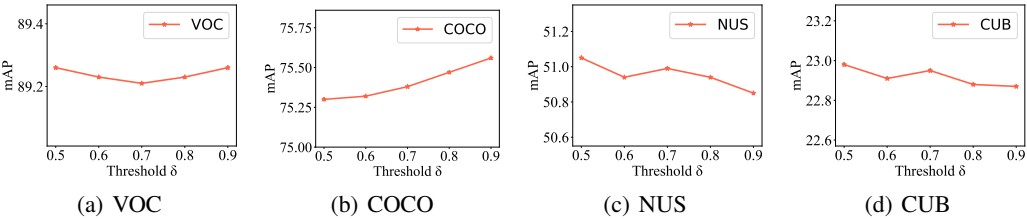

Figure 3: The performance curve as the threshold $\delta$ increases.

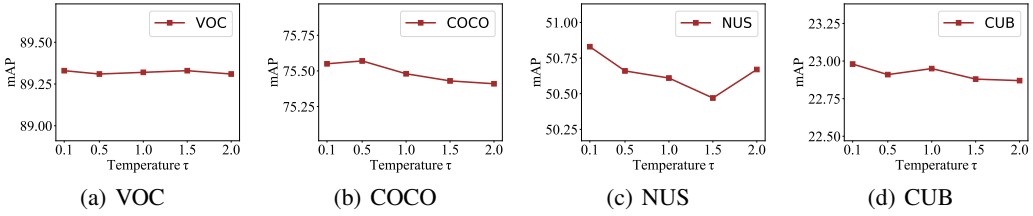

Figure 4: The performance curve as the temperature $\tau$ increases.

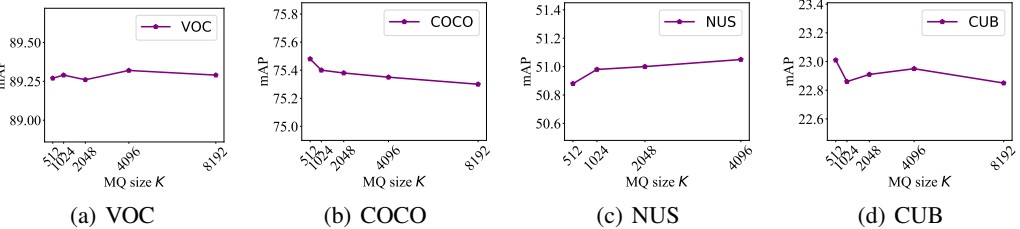

Figure 5: The performance curve as the size of memory queue $K$ increases.

beneficial for the model to identify the potential positive labels. These results convincingly validate that each component of the proposed method makes positive contribution to the final performance.

### 4.4 Parameter Sensitivity Analyses

In this section, we study the influence of three parameters, *i.e.*, threshold $\delta$, temperature $\tau$ and the size of memory queue $K$. Figure 3, Figure 4 and Figure 5 illustrate the performance curves of the proposed method as the threshold $\delta$, the temperature $\tau$ and the size of memory queue $K$ change in the range of $\{0.5, 0.6, 0.7, 0.8, 0.9\}$, $\{0.1, 0.5, 1.0, 1.5, 2.0\}$ and $\{512, 1024, 2048, 4096, 8192\}$, respectively. As shown in the figures, the performance of the proposed method is generally insensitive to these three parameters. Specifically, the largest performance gap with respect to $\delta$, $\tau$ and $K$ are respectively about $0.3\%$ on COCO, $0.2\%$ on NUS and $0.2\%$ on COCO. The results indicate that we can safely set these parameters in a large range in practice.

### 4.5 Case Studies

To disclose the mechanism behind the effectiveness of the proposed method for identifying the potential positive labels, Figure 6 visualizes some cases of attention maps on COCO. For every original image in the first (or fourth) column, we illustrate the attention maps of the single observed positive labels and identified positive labels in the next three columns. From the figures, it is interesting to observe that some small objects without annotations can be captured by the proposed method precisely, such as *potted plant* in the first row, *bowl* and *carrot* in the second row as well as *clock* and *tie* in the last row. These observations disclose that the proposed method can capture small objects by maintaining the global consistency of embeddings between the object and its intra-class objects. This further indicates LAC can significantly enhance the ability of model for identifying the potential positive labels.

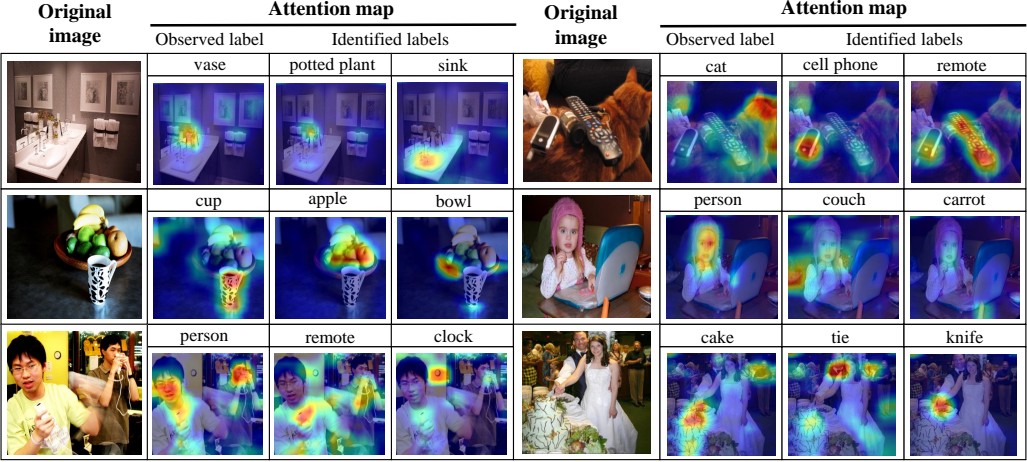

Figure 6: Visualization of attention maps on COCO.

## 5 Conclusion

The paper studies the problem of multi-label learning with single positive labels. In order to solve SPML problems, we design the PLC regularization that exploits the model outputs predicted on weakly-augmented and strongly-augmented images, and LAC regularization that leverages the manifold structure information to recover the true labeling information of the potential positive labels. By utilizing these two regularizations, the model is expected to learn a more distinctive representation, which is beneficial for identifying the potential positive labels. Extensive experimental results on multiple benchmark datasets validate the proposed method can achieve state-of-the-art performance. In the future, we plan to improve the performance of SPML by using other structure information, such as label correlations.

## Acknowledgments and Disclosure of Funding

This research was supported by the National Key R&D Program of China (2020AAA0107000), NSFC (62222605, 62076128), and Natural Science Foundation of Jiangsu Province of China (BK20211517, BK2022050029).

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
