# OpenReview forum: "Label-Aware Global Consistency for Multi-Label Learning with Single Positive Labels"
_NeurIPS.cc/2022/Conference — NeurIPS 2022 Accept_

### Official Review · Reviewer_8Jhn · 2022-07-08

**Rating:** 7
**Confidence:** 5
**Soundness:** 3 good
**Presentation:** 3 good
**Contribution:** 4 excellent

**Summary:**

This paper proposes a Label-Aware global Consistency(LAC) regularization to solve the Single Positive Multi-label Learning (SPML) problem, where each instance is only annotated with one of the positive labels related. The proposed Label-Aware global Consistency(LAC) regularization leverages the manifold structure information via encouraging the global consistency among label-wise embedding, i.e., making intra-class embedding closer while keeping away inter-class embedding.  Finally, they conduct extensive experimental results on multiple benchmark datasets, demonstrating that the proposed method can achieve superior performance.

**Questions:**

Data Report: In Table 1, the mAP of $L_{AN}$ on the dataset COCO is reported as 64.1. But in Table 2 about the ablation study on COCO, the method ‘assuming negative’ is reported as 68.5. From Page 8 Line 267-268, we can know that they are in the same setting using the same method. I wonder why there exists a 4.4% difference between them.



**Strengths And Weaknesses:**

Strength:
1. This paper proposes a novel Label-Aware global Consistency(LAC) regularization to solve the Single Positive Multi-label Learning (SPML) problem, where each instance is only annotated with one of the positive labels related.
2. This paper successfully applies ‘Clustering assumption’ in SPL and achieves good performance.
3. The visualization of the problem background, the proposed method, and the experiment results are good, which makes the reader easy to understand this work.

Weakness:
1.  The basis for saying the sentences in Page 5 Line 150-153 needs to be introduced.
2. In Page 3 Line 104, it mentions that this paper adopts the attention-based method to construct the label-wise embedding model, but in the following introduction about the proposed method, this point seems to be neglected in the proposed method, which is only related to Section 4.5 Case Study.

---

> ### Author Response · Authors · 2022-07-30
> **Thanks for great efforts on the review of this paper. We will try our best to answer all your concerns.**
>
> Thanks for great efforts on the review of this paper. We will try our best to answer all your concerns.
>
> Q1: The basis for saying the sentences in Page 5 Line 150-153 needs to be introduced.
>
> A1: $\mathcal{L}_{PLC}$ (Eq.(4)) encourages model to output the same probabilities for an unobserved label even after its corresponding example is augmented, which can be regarded as a consistency regularization in SPML case. In general, the final outputs with respect to an example can be regarded its high-level representation [1]. The previous work has proven that the effectiveness of regularization on high-level representation (final outputs) is often weaker than that on middle-level representations (e.g., instance-wise embeddings or label-wise embeddings) [1]. The observation motivates us to design the global consistency regularization to learn more distinctive feature representations.
>
> [1] Visual Attention Consistency Under Image Transforms for Multi-Label Image Classification.
>
> Q2: In Page 3 Line 104, it mentions that this paper adopts the attention-based method to construct the label-wise embedding model, but in the following introduction about the proposed method, this point seems to be neglected in the proposed method, which is only related to Section 4.5 Case Study.
>
> A2: Thanks for your suggestion. The label-wise embedding model mainly consists of a label-wise embedding encoder $f$, which generates a high-dimensional label-wise feature representation, and a classification head $h$, which makes predictions based on the label-wise feature representations. We further use a non-linear projection head $g$ to transform a high-dimensional embedding $f_j(x)$ with respect to $j$-th class into a low-dimensional embedding $z_j=g(f_j(x))$. The low-dimensional embeddings $z_j$ are used to perform label-aware global consistency regularization. The implementation details could be found in Section B in appendix. We will make our presentation clearer in the revied version.
>
> Q3: In Table 1, the mAP of on the dataset COCO is reported as 64.1. But in Table 2 about the ablation study on COCO, the method ‘assuming negative’ is reported as 68.5. From Page 8 Line 267-268, we can know that they are in the same setting using the same method. I wonder why there exists a 4.4% difference between them.
>
> A3: Thanks for your reminder. In Table 1, $\mathcal{L}\_{AN}$ represents the result of vanilla AN loss reported in the original paper [1]. Besides, we also report some results of improved version of AN loss, such as $\mathcal{L}\_{ROLE}$ + LI and  $\mathcal{L}\_{AN-LS}$ + LI (which achieves the best result on COCO in their paper). In Table 2, AN loss represents the results of our implementation, which is somewhat different from the original vanilla one on multiple aspects, such as hyper-parameter. It is noteworthy that the goal of Table 2 is to validate the effectiveness of each component of the proposed method. We will eliminate the ambiguity and make our presentation clearer in the revised version.
>
> [1] Multi-label learning from single positive labels.

---

> > ### Comment · Reviewer_8Jhn · 2022-08-08
> > **Response to Authors**
> >
> > Thanks for the authors’ feedback.

---

### Official Review · Reviewer_vE3b · 2022-07-10

**Rating:** 7
**Confidence:** 4
**Soundness:** 3 good
**Presentation:** 3 good
**Contribution:** 3 good

**Summary:**

This paper deals with the single positive multi-label learning (SPML) task. Different from vanilla multi-label learning, in SPML, only a single (e.g., the most obvious) positive label is annotated. The authors propose a label-aware global consistency loss for the task. With a two-stage optimization, the method achieves good performance on various benchmarks.


**Questions:**

The paper is good in general, and here are some questions on the implementation details.
1. Does the label embedding in Eq. 7 instance-specific? It is a bit confusing to use $K$ to index them since it is hard to infer where the label embedding comes from.
2. How about the performance without a pre-trained backbone? Does the performance of the method rely on the pre-trained backbone? The authors may consider showing the performnace without a pre-trained backbone on a certain dataset.
3. The authors may consider some more criteria to evaluate SPML. For example, to evaluate how many latent labels it could infer from the data.
4. Will the performance of the model relates to the (maximum) number of positive labels in the dataset.


-------------------------post rebuttal----------------------
I have read the response, and the authors have addressed my concerns.

**Limitations:**

The authors have adequately addressed the limitations and potential negative social impact of the work.


**Strengths And Weaknesses:**

The paper is easy to read, and the method is simple yet effective. The authors utilizes the manifold instead of the clustering assumptions in SPML, and show the advantage of the asumption based on the empirical experiments.

Here are some suggestions for the paper:
1. Figure 1 is not clear enough. By comparing the single-label and multi-label cases, the authors may first show the difference between vanilla multi-label learning and SPML. While there are some elements (such as the embeddings and regularizations) in the figure, which make the figure a bit confusing.
2. The label-wise embedding before Eq. 1 is not clear. The authors use z to denote the label embedding, but z does not exist in Eq. 1. Since the label embedding is one of the most important elements in the global consistency loss, the authors need to make it clear. Does the label embedding useful in vanilla multi-label learning (e.g., in Eq. 1-2 in the paper).
3. Since the true number of positive labels cannot be accessible in many real-world scenarios, the authors need to discuss how the proposed method can deal with this case before the experiments.

---

> ### Author Response · Authors · 2022-07-30
> **Thanks for your constructive comments. We are glad to answer all your questions.**
>
> Thanks for your constructive comments. We are glad that you considered our work “simple yet effective, good”. We are glad to answer all your questions.
>
> Q1: Figure 1 is not clear enough. By comparing the single-label and multi-label cases, the authors may first show the difference between vanilla multi-label learning and SPML. While there are some elements (such as the embeddings and regularizations) in the figure, which make the figure a bit confusing.
>
> A1: Thanks for your suggestion. In SPML, each instance is assigned with one of multiple positive labels while the other labels are unobserved. The left side in Figure 1 illustrates an example of using clustering assumption to recover the true labels of unlabeled examples in semi-supervised learning scenario (single label case). However, applying clustering assumption to solve SPML problems is not straightforward as in single label case, since it is hard to directly measure the similarity between two examples with multiple labels. The right side in Figure 1 illustrates an example of the proposed label-aware global consistency regularization for solving SPML problems, which encourages the label-wise embeddings to maintain the global consistency, i.e., maintain the clustering structure. We will make our figure and presentation clearer in the revised version.
>
> Q2: The label-wise embedding before Eq. 1 is not clear. The authors use z to denote the label embedding, but z does not exist in Eq. 1. Since the label embedding is one of the most important elements in the global consistency loss, the authors need to make it clear. Does the label embedding useful in vanilla multi-label learning?
>
> A2: Since the label-wise embedding mainly plays its role in the global consistency loss, for simplicity, we ignore the differences between the label-wise embedding and the instance-wise embedding in the content before Section 3.2. We will improve our organization and make our presentation clearer in the revised version. In general, the label-wise embeddings are also useful in vanilla multi-label learning, since it often leads model to focus on the interesting areas of the objects.
>
> Q3: Since the true number of positive labels cannot be accessible in many real-world scenarios, the authors need to discuss how the proposed method can deal with this case before the experiments.
>
> A3: Thanks for your suggestion. Unlike the previous work [1], our proposed method does not require the true number of positive labels. To perform pseudo-labels, we only need to determine the threshold $\delta$ to identify reliable pseudo labels. Our experimental results (Figure 3) show that the performance of the proposed method is insensitive to the threshold $\delta$. This means that we can safely set the parameter in a large range in practice.
>
> Q4: Does the label embedding in Eq. 7 instance-specific? It is a bit confusing to use k to index them since it is hard to infer where the label embedding comes from.
>
> A4: Thanks for your reminder. The label embedding in Eq.7 is instance-specific. For notational convenience, for the embedding $z_k$ with respect to $j$-class of instance $x_i$, we define the mapping $i=i(k), j=j(k)$ to bridge the embedding index $k$ with its original instance index $i$ and class index $j$. We will make our presentation clearer in the revised version.
>
> Q5: How about the performance without a pre-trained backbone? Does the performance of the method rely on the pre-trained backbone?
>
> A5: Thanks for your suggestion. Following the previous works [1], we use the ResNet-50 pretrained on the ImageNet to conduct all experiments. The comparisons are fair for all methods, since the pre-trained model would generally improve the performance of these methods. The study on the influence of pre-trained backbone on the performance of SPML is indeed an interesting future direction.
>
>
> Q6: The authors may consider some more criteria to evaluate SPML. For example, to evaluate how many latent labels it could infer from the data.
>
> A6: Thanks for your suggestion. We follow the previous work [1] to adopt mAP as the evaluation metric, which is the most commonly used one in multi-label literature [2][3]. The precision of identifying potential positive labels can be reflected in the final performance (mAP used in the paper) of the model, since a higher precision of identifying potential positive labels often leads to a better performance. We will show more experimental results in terms of other criteria in the future version.
>
> Q7: Will the performance of the model relates to the (maximum) number of positive labels in the dataset.
>
> A7: In general, our proposed method consistently achieves desirable performances in different benchmark datasets with different average positive labels, e.g., 1.5 for VOC and 2.9 for COCO.
>
> [1] Multi-label learning from single positive labels.
>
> [2] Multi-label image recognition with graph convolutional networks.
>
> [3] Asymmetric loss for multi-label classification.

---

### Official Review · Reviewer_fikh · 2022-07-11

**Rating:** 8
**Confidence:** 4
**Soundness:** 3 good
**Presentation:** 4 excellent
**Contribution:** 3 good

**Summary:**

The paper considers the multi-label learning with single positive labels. In such scenario, each image is only associated with a single positive label. To recover the potential positive labels, the paper first pseudo-labels the unobserved labels based on the model predictions. To boost the labeling performance, the method considers the global consistency regularization for label-wise embeddings to learn more distinctive feature representations. The experiments are conducted in multiple benchmark multi-label datasets. Experimental results show the superiority of the proposed method.

**Questions:**

Questions
1.	In many real-world scenarios, annotators may assign each image with a subset of all ground-truth labels instead of a single positive label. Can the proposed method solve this problem?
2.	In Section 4.4, Figure 3 illustrates the performance curves as the value of threshold increases. It seems that the method only achieves better performance with the increase of threshold on COCO dataset. Authors are suggested to give some detailed discussion about this phenomenon.
3.	Besides similarities among embeddings from a class label, there exists similarities among embedding from different labels. Can such similarity information help learn a good classification model? Why?


**Ethics Review Area:**

["I don’t know"]

**Limitations:**

Yes

**Strengths And Weaknesses:**

Strength
1.	The paper proposes an effective solution to SPML problems. The main contribution of this paper is to utilize the label-aware global consistency to encourage the feature representations to maintain a more compact structure. The proposed method is well motivated and easy to follow.
2.	The experiments are performed on multiple benchmark datasets to compare with state-of-art methods. The proposed method achieves impressive performances and shows significant superiority to the comparing methods. Furthermore, the ablation studies and sensitivity analyses of hyper-parameters are reported.

Weakness

1.	The experimental settings are less clear. For example, it is not introduced how to assign a single label for each image. The paper is suggested to provide some details.
2.	The label correlations are not considered in the proposed method. The label correlations have been regarded as a fundamental element for performing multi-label classification. Is it beneficial for improving the performance of SPML?
3.	In experiments, the proposed method shows large superiority on COCO than that on other datasets. Authors are suggested to provide detailed discussions about this observation.
4.	There are some language mistakes. The paper is suggested to be carefully proofread.

---

> ### Author Response · Authors · 2022-07-30
> **Thanks for your great efforts for reviewing our paper. We are glad to answer all your questions.**
>
> Thanks for your great efforts for reviewing our paper. We are glad that you considered our work “effective solution, well-motivated”. We are glad to answer all your questions.
>
> Q1: The experimental settings are less clear. For example, it is not introduced how to assign a single label for each image.
>
> A1: Thanks for your suggestion. To construct SPML datasets, we follow the previous work [1] to randomly select one of multiple positive labels for each training example. We use the same random seed used in codes of the comparing method to conduct a fair comparison.
>
> [1] Multi-label learning from single positive labels.
>
> Q2: Is the label correlation beneficial for improving the performance of SPML?
>
> A2: In general, the label correlation is beneficial for recovering the potential positive labels, which may improve the performance of SPML. However, a potential issue is how to obtain the label correlation (e.g., the co-occurrence matrix) based on SPML data (only one positive label is observed for each instance). It is really an interesting direction of the future work.
>
> Q3: In experiments, the proposed method shows large superiority on COCO than that on other datasets. Authors are suggested to provide detailed discussions about this observation.
>
> A3: Thanks for your suggestion. A possible reason is that COCO contains more labels per image (about 2.9), which provides more potential valid label-wise embeddings for performing label-aware global consistency regularization. This would lead model to learn more distinctive representations, which are beneficial to improve the performance of SPML.
>
> Q4: In many real-world scenarios, annotators may assign each image with a subset of all ground-truth labels instead of a single positive label. Can the proposed method solve this problem?
>
> A4: The proposed method can apply to the case with more than one positive label. In general, the performance of the proposed method can be improved with the increase of positive labels. It is an interesting future direction to apply our method to a more general setting.
>
> Q5: In Section 4.4, Figure 3 illustrates the performance curves as the value of threshold increases. It seems that the method only achieves better performance with the increase of threshold on COCO dataset.
>
> A5: In general, the performance of our proposed method is insensitive to the threshold (the largest performance gap is about 0.2 on COCO). On COCO dataset, it requires a large threshold to obtain more precise pseudo labels, since the number of labels per image is larger than other datasets.
>
> Q6: Besides similarities among embeddings from a class label, there exists similarities among embeddings from different labels. Can such similarity information help learn a good classification model? Why?
>
> A6: In general, the similarities among embeddings from different classes may lead to a decrease of model performance, since such similarities may introduce the ambiguity between these two classes. However, this may be an valuable problem needed to be further studied.

---

### Official Review · Reviewer_Cnop · 2022-07-11

**Rating:** 7
**Confidence:** 3
**Soundness:** 3 good
**Presentation:** 3 good
**Contribution:** 3 good

**Summary:**

This paper studies a new Multi-Label Learning (MLL) setting, i.e., the Single Positive MLL (SPML), where only one positive label is observed. The authors use AN to learn SPML. Meanwhile, they put forward Pseudo-Labeling Consistency (PLC) regularization and Label-Aware Global Consistency (LAC) regularization to address the false negative problem raising by AN. The experiments show that the proposed method (AN + PLC + LAC) achieves the best performance.

**Questions:**

Please refer to my comments above in the Strengths And Weaknesses.

**Limitations:**

The authors haven't claimed any limitations.

**Strengths And Weaknesses:**

The main strengths are as follows:
+ Although AN is not stable for SPML, the authors put forward PLC and LAC to address the false negative problem raising by AN.
+ The proposed method AN+PLC+LAC is shown to have the best performance compared with exiting SPML methods. The ablation study and case study are clear.

Besides, I also have several questions:
- PLC, i.e., Eq. (4) has the indicator function, which is not easily to be trained. I expect to see the convergence of the proposed method.
- The intra-class connection matrix I has a size of $B^2$, which is such an enormous matrix. What exactly is the size of the matrix in your experimental datasets? Is it possible to train with such an enormous matrix?
- LAC, i.e, Eq. (7) only involves the matrix I without the matrix $\bar{I}$. Where does $\bar{I}$ function?

---

> ### Author Response · Authors · 2022-07-30
> **Thanks for your appreciation of our paper. We are glad to answer all your questions.**
>
> Thanks for your appreciation of our paper. We are glad to answer all your questions.
>
> Q1: PLC, i.e., Eq. (4) has the indicator function, which is not easily to be trained. I expect to see the convergence of the proposed method.
>
> A1: The PLC regularization is used to identify the pseudo labels with high confidences. Although the indicator function is used to formulate this process for simplicity, in practice, we first identify the pseudo labels that satisfies the conditions (with confidences larger than a threshold), and then use these pseudo labels to re-train the model. This means that we do not need to derive the gradients for the indicator function, which make it easy to optimize the objective function.
>
> Q2: The intra-class connection matrix I has a size of $B^2$, which is such an enormous matrix. What exactly is the size of the matrix in your experimental datasets? Is it possible to train with such an enormous matrix?
>
> A2: In our experiments, to reduce the computational expense, we only consider label-wise embeddings with respect to potential positive labels (with confidence larger than the threshold). Since the number of potential positive labels for each instance is often small, the size of real matrix may be also relatively small, which can be regarded as a sparse form of the matrix of $B^2$ mentioned in the paper. We will make our presentation clearer in the revised paper.
>
> Q3: LAC, i.e, Eq. (7) only involves the matrix $I$ without the matrix $\bar{I}$. Where does $\bar{I}$ function?
>
> A3: Eq.(7) involves the matrix $\bar{I}$ in the denominator, which aims to minimize the similarities between the anchor embedding and the embeddings from other classes.

---

> > ### Comment · Reviewer_Cnop · 2022-08-09
> > **Read rebuttal**
> >
> > I thank the authors for their rebuttal addressing my concerns.

---

### Meta-Review · Area_Chair_rDej · 2022-08-25

**Recommendation:** Accept
**Confidence:** Certain

**Metareview:**

This paper studies the single-positive multi-label learning problem. To address this problem, the authors adopt the pseudo-labels to recover the potential positive labels and adopt the global consistency regularization for label-wise embeddings to learn more distinctive feature representations. Experimental results demonstrate the superiority of the proposal. All reviewers agree to accept this paper, so I recommend acceptance. Moreover, I still have some more suggestions: 1) The font size in Figure 3 could be larger to make the plot clear. 2) The reference format is not unified. I suggest the authors revise the reference format carefully.

**Award:**

No

---

### Decision · Program_Chairs · 2022-09-14

Accept